**METHODS AND PROTOCOLS**

# MetaPhage: an Automated Pipeline for Analyzing, Annotating, and Classifying Bacteriophages in Metagenomics Sequencing Data

Mattia Pandolfo,[a] Andrea Telatin,[b] Gioele Lazzari,[a] Evelien M. Adriaenssens,[b] Nicola Vitulo[a]

aDepartment of Biotechnology, University of Verona, Verona, Italy
bQuadram Institute Bioscience, Norwich, UK

**ABSTRACT** Phages are the most abundant biological entities on the planet, and they play an important role in controlling density, diversity, and network interactions among bacterial communities through predation and gene transfer. To date, a variety of bacteriophage identification tools have been developed that differ in the phage mining strategies used, input files requested, and results produced. However, new users attempting bacteriophage analysis can struggle to select the best methods and interpret the variety of results produced. Here, we present MetaPhage, a comprehensive reads-to-report pipeline that streamlines the use of multiple phage miners and generates an exhaustive report. The report both summarizes and visualizes the key findings and enables further exploration of key results via interactive filterable tables. The pipeline is implemented in Nextflow, a widely adopted workflow manager that enables an optimized parallelization of tasks in different locations, from local server to the cloud; this ensures reproducible results from containerized packages. MetaPhage is designed to enable scalability and reproducibility; also, it can be easily expanded to include new miners and methods as they are developed in this continuously growing field. MetaPhage is freely available under a GPL-3.0 license at https://github.com/MattiaPandolfoVR/MetaPhage.

**IMPORTANCE** Bacteriophages (viruses that infect bacteria) are the most abundant biological entities on earth and are increasingly studied as members of the resident microbiota community in many environments, from oceans to soils and the human gut. Their identification is of great importance to better understand complex bacterial dynamics and microbial ecosystem function. A variety of metagenome bacteriophage identification tools have been developed that differ in the phage mining strategies used, input files requested, and results produced. To facilitate the management and the execution of such a complex workflow, we developed MetaPhage (MP), a comprehensive reads-to-report pipeline that streamlines the use of multiple phage miners and generates an exhaustive report. The pipeline is implemented in Nextflow, a widely adopted workflow manager that enables an optimized parallelization of tasks. MetaPhage is designed to enable scalability and reproducibility and offers an installation-free, dependency-free, and conflict-free workflow execution.

**KEYWORDS** bacteriophages, NGS, bioinformatics, metagenomics, phage mining

**B**acteriophages (short phages, viruses that infect bacteria) are increasingly studied as members of the resident microbiota community in many environments, from oceans to soils and the human gut. Early studies used BLAST-based approaches and read profiling to describe the virome; this is based on the closeness of bacteriophage relationships with genomes in reference databases, leaving exceedingly large proportions of sequencing data unclassified and uncharacterized. The current gold standard

Address correspondence to Andrea Telatin, andrea.telatin@quadram.ac.uk, or Nicola Vitulo, nicola.vitulo@univr.it.

The authors declare no conflict of interest.

for phage identification includes reconstruction of (near) complete viral genomes through assembly and viral contig identification and has resulted in the following: the IMG/VR v3 database, which contains 2,332,702 distinct viral genomes or UViGs (uncultivated virus genomes) (1); and a number of human gut-associated virus databases, e.g., the Gut Virome Database (2) and the Gut Phage Database (3). Beyond assembly and identification of viral contigs, clustering in viral operational taxonomic units (vOTUs) that are aligned with the phage species level has emerged as the gold standard to reduce data set complexity (4, 5). Using 95% average nucleotide identity over 85% of the aligned fraction, the IMG/VR database contains 933,352 vOTUs. The advent of high-throughput sequencing and metagenomics has led to direct analysis and identification of the genetic material in environmental samples, overcoming the barrier of culturability. The following two different approaches are used today: sequencing of the whole metagenome followed by computational viral sequences isolation, or the physical separation of viral fractions to produce the so-called virome or metavirome.

Identification, classification, and analysis of viruses from metagenomics data sets usually involves the following five steps: (i) preprocessing of the raw reads, (ii) filtering of nontargeted reads, (iii) assembly of short sequence reads, (iv) virus identification and classification, and (v) postprocessing of the viral contigs sequences (6). Several tools exist to perform each step, but while preprocessing, filtering, and assembly can rely on gold standard software and workflows, viral identification and classification may vary greatly between different tools.

Many virus mining tools have been implemented in the last decade and differ in the targets and approaches used, which include prophage detection, homology-based identification, machine learning, and random forest or hybrid approaches that use combinations of tools (7).

A major drawback of virus metagenomic analysis is the variety of tools used in each step. Each program needs to be installed separately and may require different data formats with an extended set of dependencies that may clash in their versions. In addition, the output formats from one set of tools may not be compatible with downstream programs.

To assist the nonspecialist in the decision-making process and facilitate workflow management, we present here MetaPhage (MP), a fully automated computational pipeline for quality control, assembly, and phage detection as well as classification and quantification of these phages in metagenomics data. The pipeline is modular and enables the user to skip some of the steps and recover analysis in the event of execution errors. To guarantee scalability and reproducibility, MetaPhage was implemented in Nextflow (NF) (8), a workflow manager that uses software containers to allow easy installation. The pipeline can be run on a single computer or parallelized on an high performance computing (HPC) cluster. MetaPhage also implements a novel algorithm that delivers automatic taxonomic classification of phage contigs from the vConTACT2 (9) network graph implemented in the workflow. Results for each step of the analysis are reported on a rich and easy-to-read html report that can be opened and inspected on any web browser.

## RESULTS

**Overview of the MetaPhage workflow.** Metaphage implements a fully automated computational pipeline for phage detection, classification, and quantification of metagenomics data (Fig. 1). Several parameters are available to customize the analysis workflow, while the pipeline offers the option to skip some steps or recover analyses in the event of execution errors.

The minimal input consists of the metagenomics reads in FASTQ format. As a recommended option, the user is encouraged to also provide a metadata file containing information about the samples that can be used to customize the output. The pipeline is implemented in Nextflow, a scalable and reproducible scientific workflow manager that uses software containers to enable easy installation. A fully documented installation

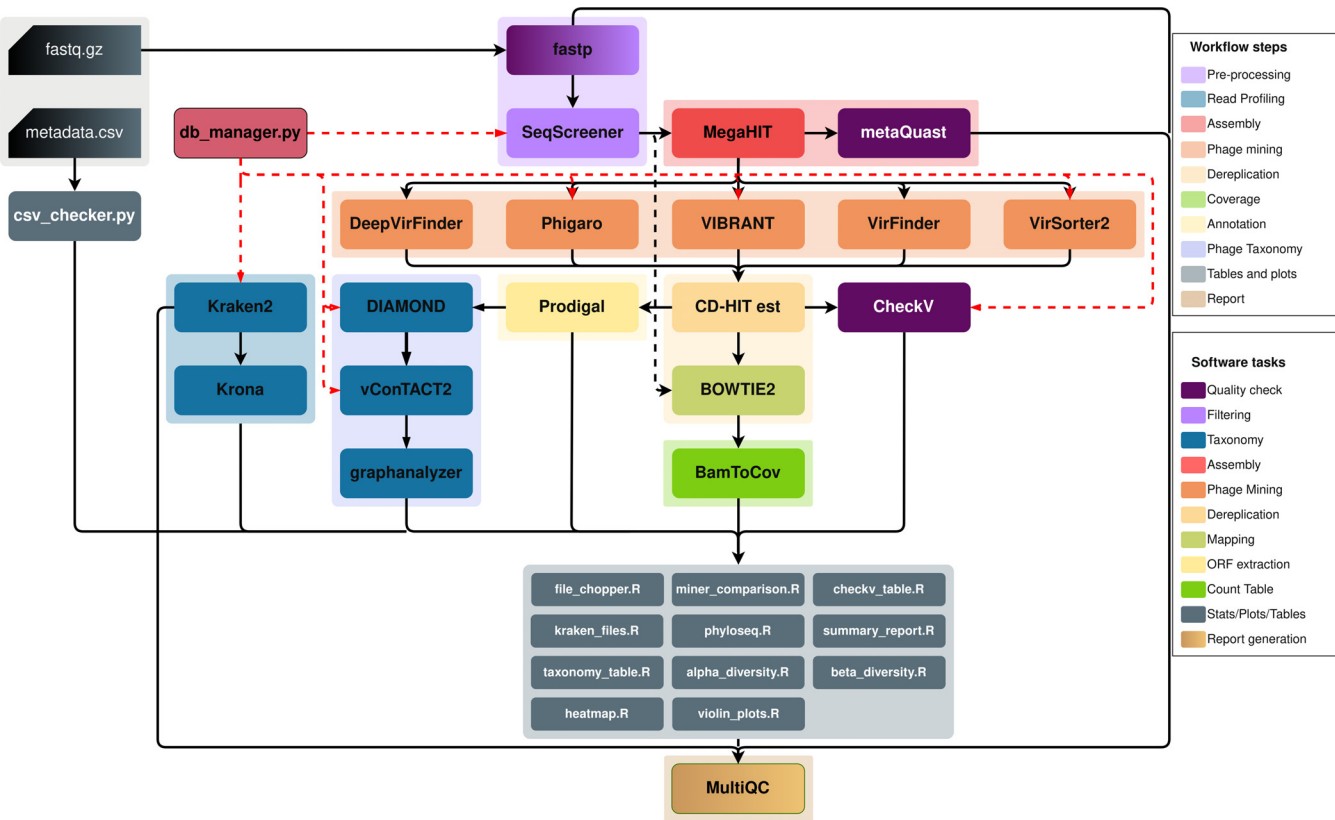

**FIG 1** MetaPhage workflow DAG chart. Input sequences can either be single-end or paired-end, while metadata are .csv files with a strict format style as described in the MetaPhage manual. By configuration file or command line, the user can decide to skip several steps or phage identification tools.

and usage guide is available at https://mattiapandolfovr.github.io/MetaPhage/. The workflow can be run on a single computer or on an HPC taking advantage of different job schedulers such as Slurm, Torque, or PBS. MP also implements a novel algorithm that allows an automatic taxonomy classification of the viral contigs using vConTACT2 clusters. Results for each step of the analysis are reported on an easy-to-read html report that can be opened and inspected on any web browser.

**Dependencies.** MetaPhage dependencies are available from the BioConda project (10) and can be exposed to the pipeline as a Conda environment, as a Docker container (11), or as a Singularity image (12), all based upon a predefined environment (YAML) distributed with the pipeline.

**Viral taxonomy classification.** vConTACT2 is the taxonomy classification tool implemented in MP. Briefly, it uses viral genomes as nodes of a monopartite network from which it derives viral clusters. Genomes are grouped based on protein content and similarity, and the resulting clusters are largely concordant with the International Committee on Taxonomy of Viruses (ICTV) viral taxa (9).

vConTACT2 splits clusters into subclusters, where clusters are an approximation of phage (sub)families, and subclusters are approximations of phage genera. For each sequence in the network, vConTACT2 assigns a label representing the clustering "status." A user can encounter several status descriptions, e.g., "Clustered" when the sequence clearly falls within a defined cluster and subcluster. More complex status classifications are also possible, e.g., "Overlap" when a sequence fits two or more clusters equally well, "Clustered/Singleton" when a subcluster consists of only one sequence, or "Outlier" when the sequence cannot be assigned to any cluster.

The network produced by vConTACT2 has viral sequences (reference genomes and vOTUs) as nodes, and its edges are weighted proportionally with the similarity between two nodes. This network can be inspected using Cytoscape (13), and this

helps the user to deduce the taxonomic context of each vOTU, including the most difficult cases. Unfortunately, repeating this manual inspection for each vOTU is very time-consuming. To automate the process, we developed a novel Python script named *graphanalyzer*, which assigns an optimal taxonomy to every vOTU following the same process as a user would do manually.

For each vOTU to be classified, *graphanalyzer* searches the corresponding node in the similarity network and retrieves the list of nodes directly connected to it (its "neighbors"), which are also in the same cluster as the vOTU (if any). The list is then ordered by the weight of the edge connecting the node to the vOTU, and the taxonomy is inherited from the first reference genome found in this sorted list. If no reference genome is found, *graphanalyzer* retrieves a new sorted list with all nodes that are simply neighbors, regardless of clustering information. If a reference genome exists within this second list, the taxonomy information is inherited from the reference with the highest weight. If no reference genome is found, the vOTU remains unclassified.

In this process, the taxonomic classification for each vOTU can be more or less confident depending on the weight assigned by vConTACT2 and the clustering status. To be more conservative, we decided to stop the inherited taxonomy at the level of subfamily if the status was "Clustered/Singleton" or "Overlap" and at the level of family if the taxonomy was inherited from a genome that was not in the same cluster. Otherwise, the taxonomy was retained until the level of genus.

The reliability of the taxonomic classification can be assessed by looking at the column "Status" and "Weight" in the taxonomy table output. The "Weight" column is the score calculated by vConTACT2 as the negative logarithm of the hypergeometric *P* value multiplied by the total number of pairwise genome comparisons. The higher this score, the higher the similarity between two genomes and so the probability they are evolutionarily related. The "Status" column refers to the label assigned by vConTACT2 to each sequence in the network. In general, a "Status" labeled as "Clustered" should be interpreted as more reliable compared to other labels, such as "Outlier" or "Overlap". The taxonomy table can be sorted according to these columns in order to rank the vOTUs based on the reliability of the taxonomic assignment.

For each vOTU, *graphanalyzer* also produces an interactive subnetwork that can be explored to retrieve extra information without the need for Cytoscape (Fig. 2). Further information on *graphanalyzer*, including its installation and usage, a description of its main algorithm and its classification levels, and a detailed user guide to the interpretation of the taxonomy table and the interactive subnetworks it produces, is freely available in its public repository at https://github.com/lazzarigioele/graphanalyzer.

**MetaPhage report.** MetaPhage results are summarized in a rich, easy-to-read html report that can be opened in any web browser. The report is organized in different sections for each step of the workflow.

The Miner Comparison section shows the number of vOTUs identified by the different phage mining tools and allows a direct comparison of the performance of each software (Fig. 3A). The Viral Contigs Distribution plot reports the length distribution on the identified viral contigs. The Summary Table section displays per-sample general statistics, such as vOTU total count, abundance, vOTU length (minimum, average, and maximum), and vOTU distribution in a searchable table. The Taxonomy Table section (Fig. 3B) is an interactive and searchable per-vOTU table, displaying taxonomy information and host association information (retrieved using the information contained in the INPHARED database [14]). The taxonomy is automatically assigned using the *graphanalyzer* tool, and it stops at the genus level if available. For each vOTU, it is also possible to visualize the corresponding subnetwork generated by *graphanalyzer*, together with its nucleotide consensus sequence, predicted protein sequences (FASTA format), and coordinates files (GFF3 format). The table allows the user to query for any combination of phage miners to restrict the search to only the vOTU of interest.

The vOTUs Distribution (Fig. 3A) section contains links to the generated heatmap

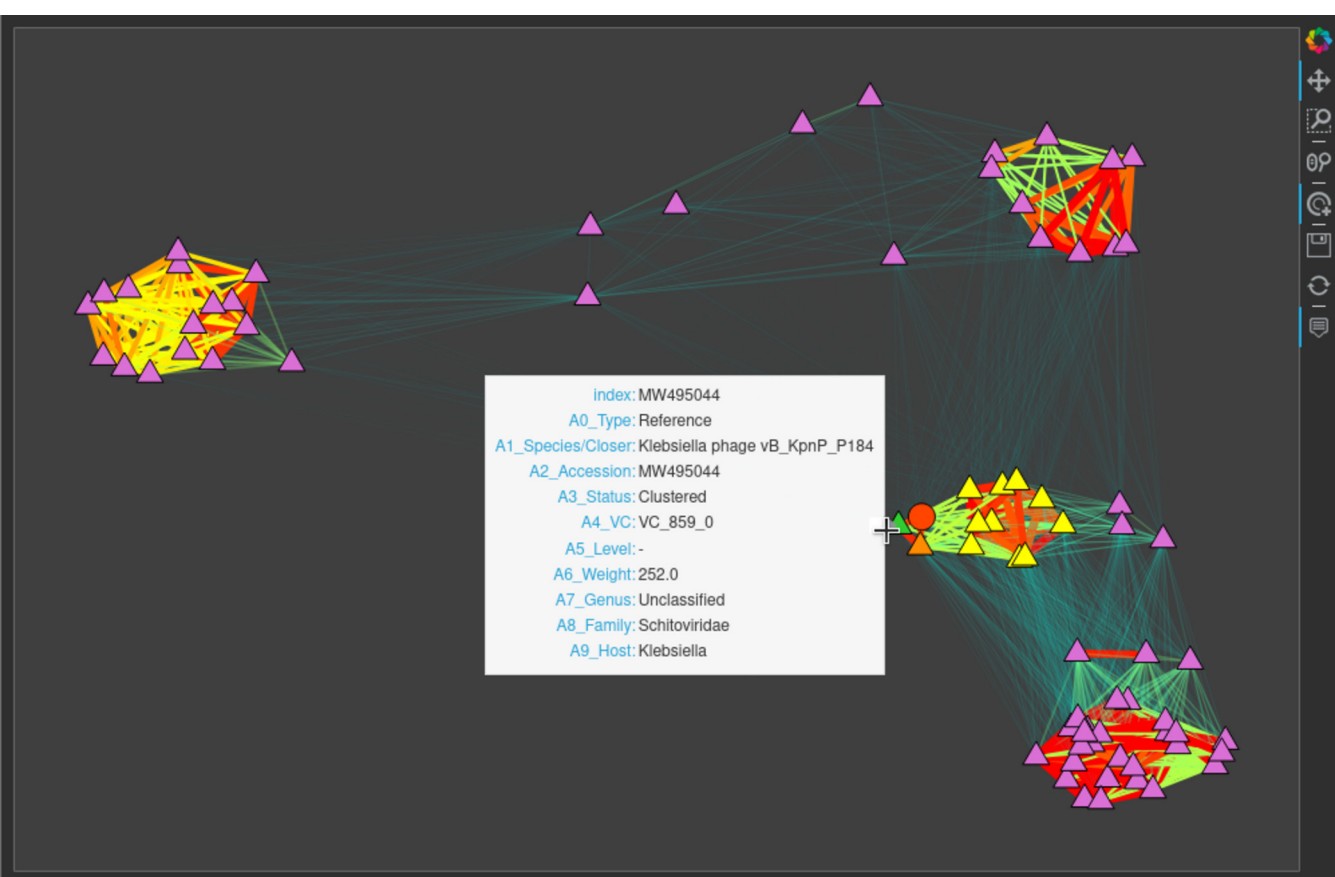

**FIG 2** Interactive subgraph produced by *graphanalyzer* for a vOTU. Triangles and circles represent reference genomes and vOTU, respectively. This vOTU and the reference genome from which it inherits the taxonomy are depicted in red and green, respectively. Orange nodes are subclustered together with this vOTU, while yellow nodes are only clustered together and belong to a different subcluster. The width and color of edges are proportional to the similarity between two nodes, from thin transparent aquamarine (weaker similarity) to thick opaque red (strongest similarity) shading from green, yellow, and orange in-between. Nodes are positioned approximately respecting the similarity between them, and this tends to make clusters visible at first sight. When opened with a browser, this subgraph is interactive; the user can zoom and drag it and can hover over a node with the mouse to show its properties, like taxonomy classification or cluster type.

and violin plots, both interactive plots that give information about vOTU abundance (log count) and distribution across the samples.

The report also provides alpha and beta diversity metrics in the corresponding report sections to describe within-sample complexity and between-sample diversity. In particular, alpha diversity is calculated using different metrics, such as observed, Shannon, Abundance-based coverage estimators, Simpson, and Fisher, while beta diversity is calculated using Bray-Curtis and Jaccard indices.

Several important files that are generated during the analysis and are useful for downstream analysis are also directly linked within the report for easy and fast access, in the Relevant Files section (Fig. 3C). In particular, the vOTU count table reports the abundance of each vOTU in each sample, the taxonomy table, multi FASTA files containing the vOTU nucleotide and protein sequences, and the coordinate of the predicted proteins on the corresponding contigs in gff3 format. Moreover, a phyloseq object is also provided with both raw and cumulative sum scaling (css) normalization counts. phyloseq is an R package to import, store, analyze, and graphically display complex OTU-clustered high-throughput phylogenetic sequencing data (15). The package implements advanced/flexible graphic systems and leverages many of the tools available in R for ecology and phylogenetic analysis for downstream analysis.

The report also contains the output of the taxonomic classification performed using kraken2 (16), providing both an interactive bar plot (17) summarizing the results of the

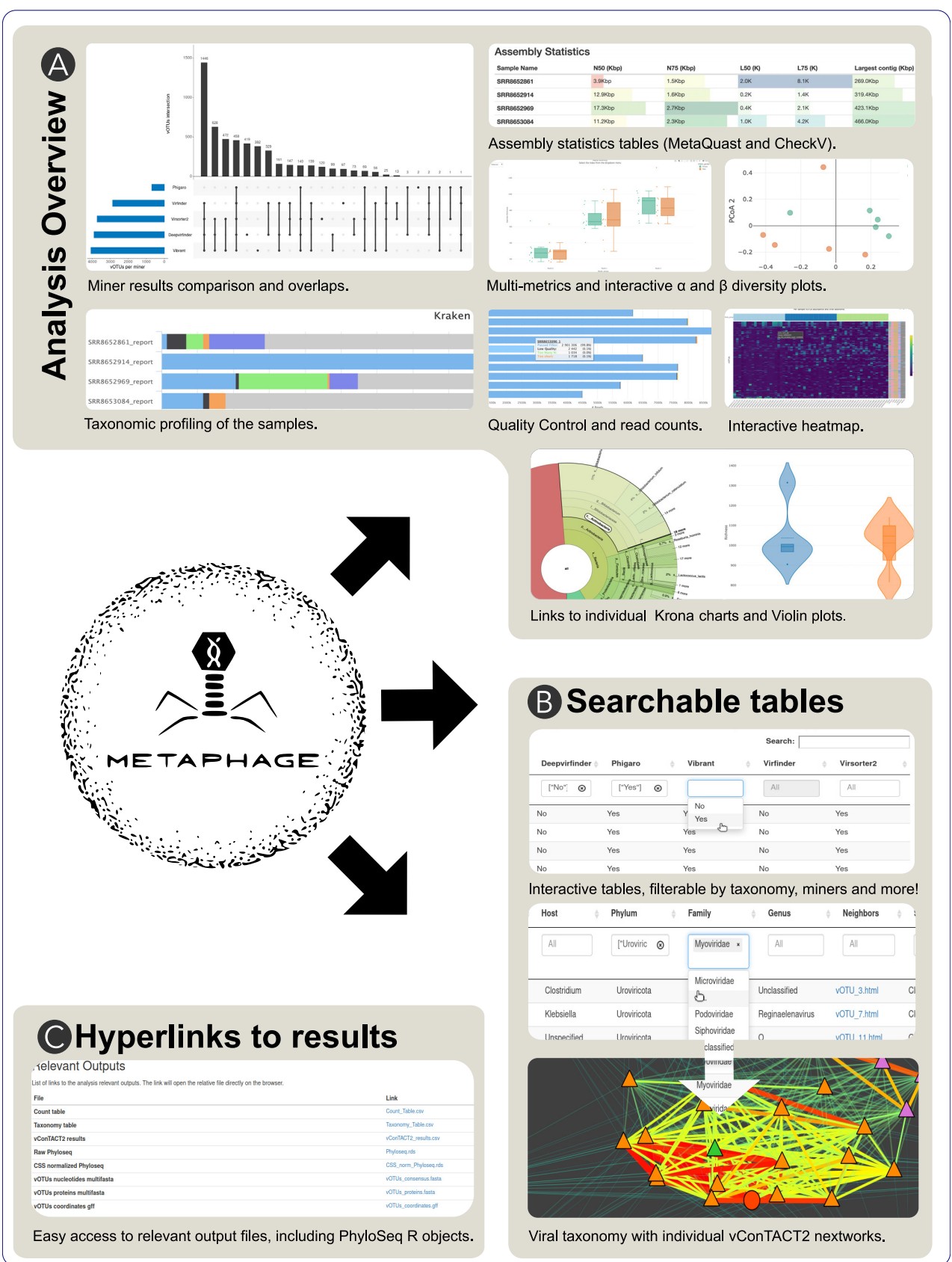

**FIG 3** MetaPhage produces an html report with multiple sections, which can be divided in three main categories as follows: analysis overview, searchable tables, and hyperlinks to results. (A) Analysis overview includes panes to inspect the overall quality of the reads (*fastp*) and taxonomic

taxonomic classification on the whole data set and both the single Kraken2 and Krona pie chart output file for each sample (Fig. 3A).

Finally, the last part of the report provides general statistics about read quality check/preprocessing and assembly metrics (calculated using the metaQuast tool [18]).

**Validation and testing.** MetaPhage performance was tested by analyzing a DNA virome derived from virus-like particles (VLP) and a shotgun metagenome data set from a published study (19). The authors analyzed 20 healthy infant stool samples, collected at 0 to 4 days after birth and at 1 and 4 months of life, in order to investigate the origin of the viral population in the gut. The authors reported 2,552 viral contigs, 1,029 of which were classified as bacteriophages containing more than 10 open reading frames (ORFs).

Our MetaPhage analysis, run considering the minimum vOTU length of 3,000 as described in reference 19, found 5,284 vOTUs (2,507 of which contained more than 10 ORFs) in the DNA virome data set. Twenty-seven percent of the identified vOTUs were predicted by either VirFinder (20), VIBRANT (21), VirSorter2 (22), or DeepVirFinder (23), while 8.6% were predicted by all of the miners. VIBRANT was the miner that predicted the higher number of vOTU (4,078). In the shotgun metagenomics data set, we predicted 14,274 vOTUs (4,666 of which contained more than 10 ORFs). A total of 24.8% of the identified vOTUs were predicted only by DeepVirFinder and VIBRANT, while only 2.3% were predicted by all of the five miners. DeepVirFinder was the miner that predicted the overall higher number of vOTU (10,508). Details on the number of vOTUs predicted by the different miners on the two data sets can be found as at https://doi .org/10.6084/m9.figshare.20449644. Both the data sets were classified using vConTACT2, and the taxonomy assignment was done by means of our *graphanalyzer* tool as described in the Materials and Methods and Results sections. In the DNA data set, the taxonomy classification allowed us to identify 3 phyla (1,943 vOTUs), 3 classes (1,943 vOTUs), 4 orders (1,954 vOTUs), 9 families (1,894 vOTUs), and 32 genera (87 vOTUs). In the shotgun data set, we identified 3 phyla (2,376 vOTUs), 3 classes (2,376 vOTUs), 4 orders (2,376 vOTUs), 9 families (2,321 vOTUs), and 19 species (47 vOTUs).

Bacteriophage classification results, collapsed at the family taxonomic rank and reads per million (RPM) normalized, are shown in Fig. 4A. These results are largely in concordance with those of the original authors, with differences attributed to an increased number of phage genome sequences available from public databases for taxonomic assignments and an increase in the number of phage families defined. Results are also confirmed looking at the alpha diversity (Fig. 4B), which shows a lower richness and diversity at month 0 compared to month 1 and month 4. As phage taxonomy changes over time, MetaPhage will be able to easily adapt and incorporate new taxa using updated versions of the core databases.

The full report generated by MetaPhage for the VLP data set is available for download at https://doi.org/10.6084/m9.figshare.20424705.v1.

**Time consumption.** The validation testing was run for both the data sets on Cineca's Galileo100 HPC cluster (https://www.hpc.cineca.it/hardware/galileo100) using SLURM as a job scheduler and requesting a variable number of resourcing ranging from 4 to 48 cores and from 8 GB to 64 GB of RAM depending on the type of process.

The DNA virome data set analysis (60 samples) was completed in 23 h. The sum of each single process real time was around 60 h (without considering the jobs running in parallel). On average, per sample, DeepVirFinder processes ran in 6 min, Phigaro (24) in 1 min, VIBRANT in 2 min, VirFinder in 1 min, and VirSorter2 in 23 min. DIAMOND (25), vConTACT2, and *graphanalyzer* together took 3 h 43 min to complete.

The shotgun metagenome data set (59 samples) was run in 37 h. The sum of each

**FIG 3** Legend (Continued)
composition of the whole sample (*Kraken2*), including interactive plots for each sample (*Krona*), the assembly metrics (*metaQuast*), and custom plots specific to viral diversity and produced with R by the pipeline (alpha and beta diversity, heatmap, violin plots). (B) The searchable tables include a summary of the taxonomic analysis of the viral OTUs (vOTUs) as performed (vConTACT2), and custom filters can be added to exclude some miners or to restrict the search to specific phage clusters in the network. For each vOTU, a link to the individual subnetwork (as performed by the *graphanalyzer* script). (C) A dedicated section reports links to the main files produced by the pipeline for downstream analyses, including the raw counts table, the taxonomy table, and phyloseq objects for downstream analyses in R.

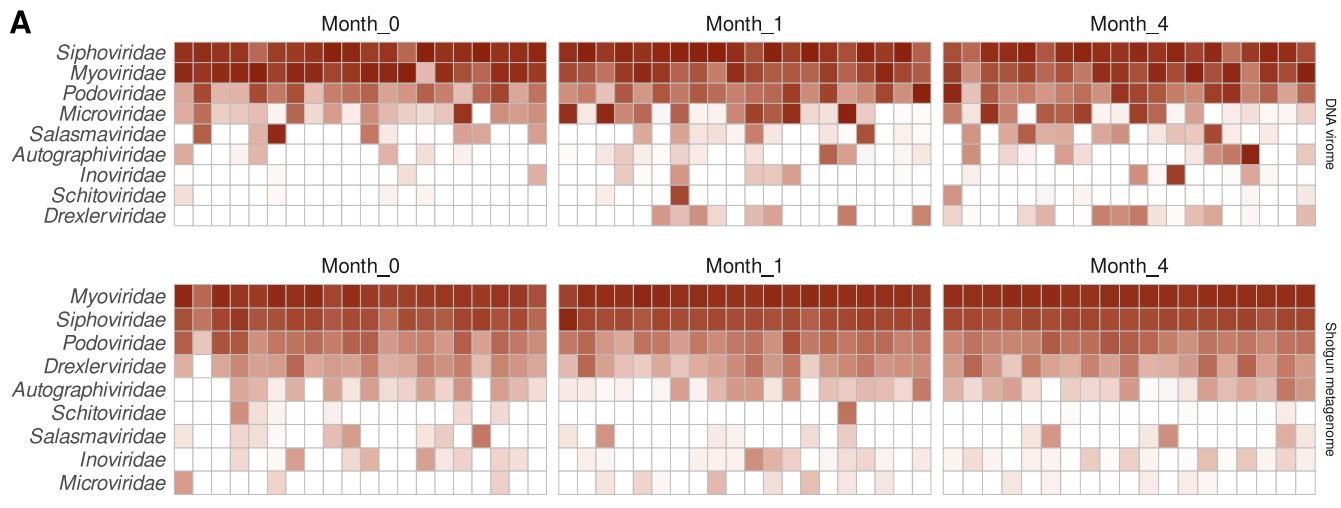

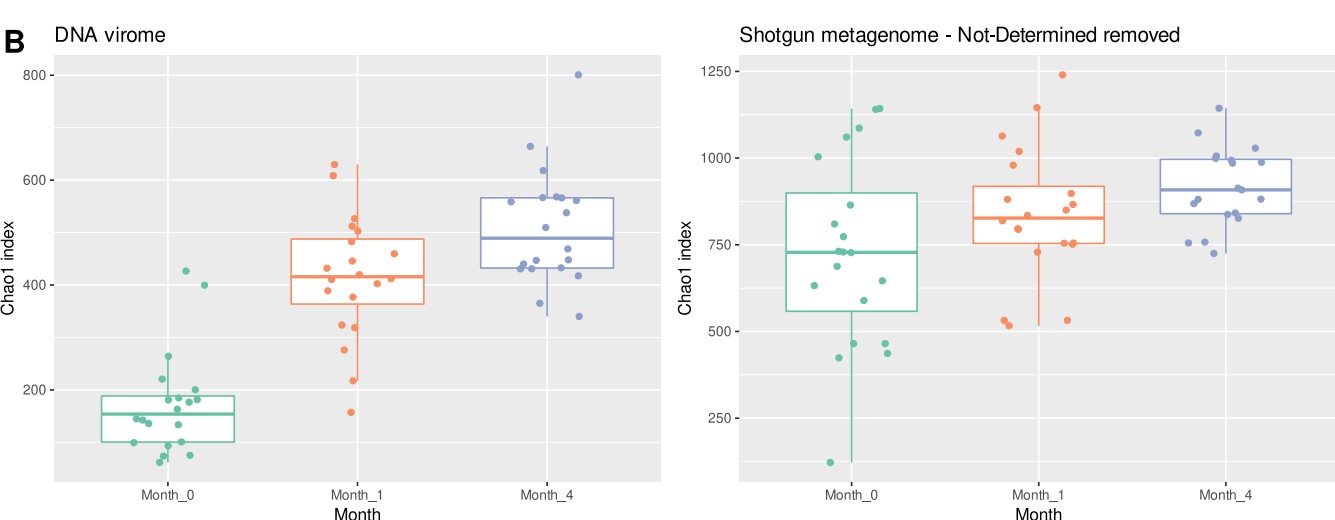

**FIG 4** (A) Heatmaps of the RPM (reads per million) abundances of predicted viral families across each sample in the two data sets tested. (B) Alpha diversity stratified by sampling time based on Chao1 metrics. The vOTUs unclassified at Phylum level were removed. Low quality vOTU labeled as "not-determined" by CheckV quality assessment were also removed in the shotgun metagenome data set.

single process real time was slightly more than 171 h. On average, per sample, DeepVirFinder processes ran in 32 min, Phigaro in 5 min, VIBRANT in 10 min, VirFinder in 10 min, and VirSorter2 in 1 h and 32 min. DIAMOND, vConTACT2, and *graphanalyzer* together took 4 h and 46 min to complete.

## DISCUSSION

MetaPhage is a modular and automated tool to detect, annotate, and analyze bacteriophages present in a variety of different metagenomic data sets (DNA or RNA viromes, shotgun metagenomes), allowing the user to select and combine four phage miners for the analyses. Detection of viral contigs by more than one method is recommended to obtain comprehensive results since different bacteriophage miners based on different theoretical premises can complement each other (26). This recommendation was also implemented in the What the Phage pipeline (27), which implemented multiple tools for phage prediction in one workflow. While MetaPhage uses a similar approach to using multiple phage detection tools, its implementation starts from raw

reads and provides a taxonomic annotation for each assembled phage sequence as part of a rich interactive report made up of useful plots and tables that can be opened on any browser and downloaded directly on the machine.

We used previously published data sets of shotgun and VLP metagenomes of infants after birth (19) for the testing and validation of MetaPhage as described in the results. This allowed us to compare our results with the original data analysis but also highlight the consequences of the use of different input data. When comparing our results with those of Liang and colleagues (19), we saw high concordance in the number of vOTUs predicted and a pattern of increased richness from month 0 to 4 (Fig. 4). In the taxonomic assignments of the phages, MetaPhage provided a higher resolution because of an update to phage taxonomy resulting in the creation of more families (28). Our results also showed clear differences in the detection of vOTUs from the bulk metagenomes and VLP extracts. Quality assessment with CheckV (29) showed that the large difference in number of vOTUs predicted was due to predictions of more partial and low-quality genomes, revealing 569 high-quality or complete genomes in the VLP data set and 397 in the shotgun metagenomes. We hypothesize that the excess predictions in the shotgun metagenomes are due to a combination of low sequence coverage of the rare phages in the microbiome (resulting in low-quality genomes) and an increase in false positives (e.g., plasmids, bacterial fragments identified as viral). This finding nicely illustrates the ongoing balancing act between recall and precision in virus prediction that is also discussed by the virus miner developers, and as such, we urge the users to be mindful of the biases inherent to virome analyses. Additionally, as expected and concordant with findings by Liang and colleagues, more proviruses were predicted in the shotgun metagenome than the VLP data (757 versus 113).

From a computational perspective, MetaPhage is a scalable pipeline that can be run on any platform that supports Conda, Singularity, or Docker. The defined dependencies between the various steps of the calculations and different intermediate level result files allow the user to rerun the pipeline and restart the analyses from the last concluded step if needed. MetaPhage makes it easy to perform a complex analysis of bacteriophages in metagenomics, yielding reliable results that facilitate understanding and visualization of the composition and structure of viral communities.

In conclusion, MetaPhage is an automated pipeline that performs phage-mining and viral taxonomy classification in metagenomics data. In a single run, MetaPhage provides quality control, read profiling, assembly, phage detection, phage classification, and quantification. MetaPhage's key findings are summarized in an exhaustive html report, which enables further exploration via interactive plots and filtering tables. MetaPhage has an easy installation process and makes the pipeline useful for non-bioinformatician users. MetaPhage implements a new algorithm for the automatic taxonomy classification of the viral contigs using the output of vConTACT2.

## MATERIALS AND METHODS

**Overview.** The pipeline was implemented in Nextflow (NF) [v21.04.0], a portable, scalable, and parallelizable workflow manager. MetaPhage (MP) includes five different phage-mining tools that identify viral or phage sequences in prokaryotic metagenome data, together with 14 other tools for quality control, assembly, clustering, annotation, and classification as shown in Fig. 1. These tools are integrated into MP using Singularity (v3.7.1) (12), a container platform that packages software in a single file (i.e., container), which is portable and reproducible; this approach frees the user from the need to install software and dependencies, which may be challenging for beginners in bioinformatics. Therefore, MP offers an installation-free, dependency-free, and conflict-free workflow execution, regardless of the operating system.

The six mandatory databases needed for analysis are automatically downloaded and stored by the pipeline on the first execution. Moreover, users can define different custom databases through the MP config file. Almost every pipeline step can be skipped, and many options are fully customizable by the user through the MP config file. If no custom config is provided, the pipeline runs with default options. The details of the main steps are reported below.

**Database download.** MP relies on several external databases as follows: (i) *Phix*, which removes the calibration control of sequencing runs during the preprocessing step; (ii) *MiniKraken2_V1* as default or the Kraken2 database used in the read classification step (available at https://benlangmead.github.io/aws-indexes/k2); (iii) VIBRANT (21); (iv) Phigaro (24); (v) VirSorter2 (22) databases, which contain all of the dependencies needed by the relative tools; and (vi) the INPHARED (14) database which provides a

collection of curated bacteriophage genomes from GenBank and related metrics, together with several input files useful in bioinformatic pipelines.

**Preprocessing and sequence manipulation.** MP takes single-end/paired-end Illumina fastq files as input. The first step is to use the fastp program (v0.20.1) (30) to remove user-defined adapter sequences (TrueSeq adapters by default) and trim raw reads using a sliding window approach. The -5 and -3 options are used to move from 5′ to tail and from 3′ to head of each sequence, trimming where quality falls below a user-defined threshold set with the –*cut_mean_quality* option (15 by default). If no adapter sequences are provided (via −*adapter_sequence* and –*adapter_sequence_R2* fastp options, respectively), the tool tries to automatically detect the correct adapters for the input sequences. The *SeqScreener* module from HTStream package (v1.3.3) (31) (reimplemented as part of HTStream, https://github.com/s4hts/HTStream) is then run to discard contaminants mapping the *phiX174* genome (default) or any user-defined genome, and the resulting phix-depleted and preprocessed reads are sent to the next stage, which runs in parallel (see Read classification and Genome assembly sections below). Manipulation of FASTA files is done via SeqFu (v1.9.6) (32).

**Read classification.** Microbial taxonomy classification of all reads in the sample is done by the Kraken2 tool (v2.1.2) (16). To enable the user to perform further downstream analysis of microbial classification, Kraken2 is run with the option –*report-zero-counts*, which displays all taxa, even those not assigned to any reads. The Kraken2 output is then passed to Krona (v2.8) (17), which summarizes the results, creating interactive html pie charts. This step provides a general overview of the microbial composition within the samples, including any unexpected or unwanted sequence that may be potential contaminants. Notice that it is not tailored for identification of phages but to provide an ecological context that can be useful for better interpretation of the results.

**Genome assembly.** Reads are assembled using the MEGAHIT assembler (v1.2.9) (33) with default parameters. Assembled scaffolds are evaluated on their quality through the Assembly quality check step, which uses metaQuast (v5.0.2) (18) as a tool to evaluate and compare metagenome assemblies based on alignments to close references. Since reference genomes are not known *a priori*, metaQuast uses BLASTN to identify the metagenome content aligning the contigs to the SILVA (34) 16s rRNA database. Based on these results, it automatically downloads the reference genomes that are used for the quality assessment. To find rRNA genes the –*rna-finding* option is used. The resulting report includes several statistics per assembly, including contigs cumulative length, $N_{50}$, and GC content; these are plotted in an html report file, with interactive graphs. Concurrently, the assembled scaffolds outputted by MegaHIT are passed to the Phage mining step, for identification of bacteriophage genomes.

**Phage mining.** MP exploits the following five phage mining tools running in parallel to identify viral sequences in the contigs/scaffolds: DeepVirFinder (v1.0) (23), VIBRANT (v1.2.0), Phigaro (v2.3.0), VirSorter (v2.2.3), and a parallelized version of VirFinder (v1.1) (20). The first four tools can be run with the default database available with the program or with a user-provided database. DeepVirFinder runs with default parameters. VIBRANT runs with default parameters (minimum scaffold length requirement set at 1,000 bp and minimum number of open reading frames (ORFs) per scaffold set at 4). The option –*virome* increases sensitivity for virome data sets by removing nonviral scaffolds and is used when the MP option –*virome-dataset* is set to true by the user. Phigaro runs with default parameters in basic mode, since GC quantity levels may vary depending on the input data set (see reference 24 for details). VirSorter2 runs with default parameters, using a minimum contig length (command option –*min-length*) of 0 by default, which the user can change via the –*virsorter2_min-length* parameter setting it in the configuration file. The –*include-groups* option is set by default to double-stranded DNA (dsDNA) and single-stranded DNA (ssDNA) viral groups, but the user can change it to different viral groups (e.g., RNA viruses) via the –*virsorter2_include_groups* parameter in the pipeline. The identified viral contig sequences are then sent to the Dereplication step.

**Dereplication and viral contigs comparison.** The dereplication process consists of clustering viral contigs according to a nucleotide similarity threshold. Viral contigs are dereplicated using CD-HIT-EST (v4.8.1) (35) with a sequence identity threshold of 0.95, word size of 9, and alignment coverage of 0.85. This step makes use of the "accurate mode" (-*g*: 1) by which sequences are clustered to the most similar cluster that meets the threshold. CheckV (v 0.9.0) (29) is then used to assess the consensus viral contigs outputted by CD-HIT-EST (from now on referred to as vOTUs). The tool runs with the end_to_end command option in order to run its full pipeline using default parameters. These steps consist of the identification of host contamination for integrated proviruses, the estimation of genome fragment completeness, and the closed genome identification. To compare results from different phage-mining tools, an R script based on the UpSet R library (v1.4.0) (36) is used to generate a comparison plot (more details are provided in the Results section). This graph provides a general overview of how many vOTUs are identified by each miner (i.e., after dereplication) and how many are mined by two or more tools. vOTUs are the input for the following concurrently running steps in the pipeline, i.e., Mapping and Annotation.

**Mapping and count table construction.** Bowtie2 (v2.4.2) (37) is used to quantify vOTUs in all phix-depleted, preprocessed read sets, mapping them back and processing the results with SAMtools (v1.11) (38). The resulting alignment is sent to the Count table construction step, which makes use of the *bam-countrefs* module (default parameters) of the BamToCov tool (v2.0.4) (39) to create a count table. The module is run with option –*multiqc* to output the table in the correct format, and the resulting table presents the raw reads count with no normalization applied.

**Annotation and phage taxonomy.** Concurrently with the Mapping step, Prodigal (v2.6.3) (40), a prokaryotic and viral gene recognition tool, is used in metagenome mode to predict viral ORFs from the vOTUs. This tool makes use of an unsupervised machine learning algorithm that automatically learns the genome properties (start codon, ribosomal binding site motifs, coding statistics) without a training data

set. The metagenome mode option (–*p meta*) is mandatory when working with mixed samples, such as the assembly of a metagenome.

The resulting protein sequences are then passed to the viral taxonomy step, which uses DIAMOND (v0.9.14) (25) to align them against the reference database. The alignment is then passed to vConTACT2 (v0.9.19) (9), a tool that performs network-based clustering of viral sequences and specifically designed to work on metagenomics data. vConTACT2 uses whole-genome gene-sharing profiles between a reference database of curated and annotated bacteriophages and the user's viral sequences to build networks for phage taxonomy. One of the key steps in the vConTACT2 pipeline is an all-versus-all predicted protein comparison that generates clusters of protein families. The reference database and user's viral proteins are merged into a single file to perform an all-versus-all similarity search. Depending on the size of the reference database, this step can require a long time as suggested by the authors of the tool. In addition, the network is rebuilt every time new data are added, extending the time required even further.

We reason that if the reference database does not change, then the all-versus-all protein comparison of the reference needs to be done just once, as the results will not change. In order to speed up the process, we decided to split the all-versus-all comparison to the reference database from the user viral proteins search, running vConTACT2 using *vConTACT2_proteins.faa* and *vConTACT2_gene_to_genome.csv*, two files contained in the INPHARED database. For a detailed description of the approach see the Custom vConTACT2 script section available at https://doi.org/10.6084/m9.figshare.20449644.

The similarity network and the clusters computed by vConTACT2 are then processed by *graphanalyzer* (v1.2.1), a novel script that we developed to automatically inherit the taxonomy of each vOTU from its closest reference genome. This reference is chosen by looking for connected genomes inside the network and prioritizing those included in the same vConTACT2-defined cluster. This saves time and effort, since vConTACT2 does not classify a vOTU directly but instead would leave the classification up to the user's extensive manual inspection. Moreover, *graphanalyzer* produces a summarizing taxonomy table (.csv file) that is passed to the *Plots&Report* step, as well as an interactive subgraph for each vOTU, which enables the user to explore its taxonomic context. The tool was written in Python and is freely available at https://github.com/lazzarigioele/graphanalyzer. Further details of its algorithm are discussed in the Results.

**Plots and report generation.** To produce interactive plots and tables, several R scripts make use of the following: the vOTUs count table produced by BamToCov, the taxonomy table produced by *graphanalyzer*, and the opportunely formatted metadata (.csv) file (see MetaPhage output section in Results for in-detail explanation). These scripts make use of different R libraries, such as plotly, phyloseq (15), metagenomeSeq (41), tidyverse (42), seqinr (43), heatmaply (44), DT (45), and Biostrings (46). The pipeline steps are summarized via MultiQC (47) software, which produces a comprehensive interactive report, available via web browser. This report integrates several software output reports (fastp, Kraken2, metaQuast), together with the custom interactive plots (more details in the Results).

**Availability.** MetaPhage is available under a GPL-3.0 license at https://github.com/MattiaPandolfoVR/MetaPhage while *graphanalyzer* is available under the same license at https://github.com/lazzarigioele/graphanalyzer. MetaPhage documentation is available online at https://mattiapandolfovr.github.io/MetaPhage.

## ACKNOWLEDGMENTS

We thank Judith Pell for scientific writing services.

This work was supported by the 3S_4H Project, Safe, Smart, Sustainable food for Health, ID 10065201; FESR Regione 2017, POR FESR 2014–2020; and ISCRA Italian SuperComputing Resource Allocation projects (IscraC, IsC92_MPhage) founded by CINECA. E.M.A. and A.T. have been supported by BBSRC Institute Strategic Program Gut Microbes and Health BB/R012490/1 and its constituent projects BBS/E/F/000PR10353 and BBS/E/F/000PR10355. A.T. and N.V. established the collaboration thanks to a BBSRC Flexible Talent Mobility Account scheme award (BB/R506552/1). The pipeline has been tested on CLIMB-BIG-DATA computing infrastructure, funded by the UK's Medical Research Council through grant MR/T030062/1.

M.P., G.L., and A.T. developed the MetaPhage pipeline, working together with N.V. and E.M.A. to design MetaPhage and write the first draft. N.V., A.T., and E.M.A. supervised the project and did the critical reading, correction, and revision of the manuscript, together with general adjustments for the metagenomic workflow.

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
